# A Fully Differential Analog Front-End for Signal Processing from EMG Sensor in 28 nm FDSOI Technology

**DOI:** 10.3390/s23073422

**Published:** 2023-03-24

**Authors:** Vilem Kledrowetz, Roman Prokop, Lukas Fujcik, Jiri Haze

**Affiliations:** Department of Microelectronics, Brno University of Technology, Technicka 3058/10, 61600 Brno, Czech Republic

**Keywords:** common-mode rejection ratio (*CMRR*), fully differential difference amplifier (FDDA), driven-right-leg circuit, active ground circuit, fully depleted silicon on insulator (FDSOI), electromyography (EMG)

## Abstract

This paper presents a novel analog front-end for EMG sensor signal processing powered by 1 V. Such a low supply voltage requires specific design steps enabled using the 28 nm fully depleted silicon on insulator (FDSOI) technology from STMicroelectronics. An active ground circuit is implemented to keep the input common-mode voltage close to the analog ground and to minimize external interference. The amplifier circuit comprises an input instrumentation amplifier (INA) and a programmable-gain amplifier (PGA). Both are implemented in a fully differential topology. The actual performance of the circuit is analyzed using the corner and Monte Carlo analyses that comprise fifth-hundred samples for the global and local process variations. The proposed circuit achieves a high common-mode rejection ratio (*CMRR*) of 105.5 dB and a high input impedance of 11 GΩ with a chip area of 0.09 mm${}^{2}$.

## 1. Introduction

Medical diagnostic procedures such as electromyography (EMG), electrocardiography (ECG), electroencephalography (EEG), or electrooculography (EOG) have significant clinical importance in the diagnosis of numbness, muscle weakness (EMG), many common heart problems (ECG), epilepsy, brain disorders (EEG), etc. These methods have been known for a long time; however, their development is continually progressing, the same as their sensing and processing development.

Biomedical information processing comprises techniques that apply mathematical tools to extract crucial diagnostic information from biomedical data. The main steps of a typical biomedical measurement and processing system are shown in Figure 1.

The first step is to identify the relevant physical properties of the biomedical system, followed by measuring using suitable sensors. The next step is preprocessing and filtering. The preprocessing block comprises a low-noise instrumentation amplifier (INA) and a programmable-gain amplifier (PGA). Together with a filter, it constitutes an analog front-end component (AFE). Preprocessing by the INA is necessary because the measured electric signal is weak and must be increased in amplitude so it can be further processed. Magnifying of the amplitude is the essential function of the AFE, which must meet many specific requirements. These are listed below:high input impedance to provide minimal loading of the measured signal;high open-loop gain;variable amplification selective to the physiological signal;high rejection of superimposed noise and interference signals;high common-mode rejection ratio (*CMRR*) and guaranteeing the safety of the patient;low power consumption.

The high-pass (HPF) and low-pass (LPF) filters aim to eliminate interference signals such as electrode half-cell potentials or offset of the INA and reduce the noise level by limiting the amplifier bandwidth. Filtering is followed by converting an analog signal into a digital signal, which enables further digital signal processing (DSP). The delta-sigma analog-to-digital converter (ADC) is currently one the most popular types of ADC architecture in biomedical signal processing, owing to its power efficiency and high resolution [1,2,3]. It is also possible to implement only a delta-sigma modulator on a wearable sensor integrated circuit (IC) when a stream of single bits from the modulator´s output (PDM) is transmitted via wireless communication to a computer [4,5].

Many authors have introduced AFE for ECG or EEG signal processing, in which requirements for signal ranges in frequency and amplitude are lower than in EMG, as can be seen in Table 1.

Parameters related to the frequency spectrum of ECG or EEG signals enable achieving a very low power consumption of AFE in hundreds of nW [7,8,9]. Some articles deal only with the design of a differential difference amplifier (DDA) as an attractive replacement for a classical INA composed of three operational amplifiers [10,11,12,13]. For example, the authors of [13] introduced a DDA suitable for ECG signal processing with a power consumption of only 313 nW.

As can be seen from Table 1, the EMG signal ranges in frequency from 20 Hz to 2 kHz depending on the type of examination. Moreover, setting the bandwidth of the AFE to above 20 Hz can greatly reduce the skin potentials and motion artifacts [14]. Signal amplitudes range from tens of µV to approx. 10 mV, depending on the type of signal and electrodes used. Thus, EMG amplifiers must have a wider frequency response than ECG amplifiers.

This paper aims to design an AFE suitable for EMG signal processing with the low power supply voltage VDD = 1 V. Such low voltage considerably limits the design options for each analog block when it is difficult to use cascodes. For this reason, FDSOI 28 nm technology from STMicroelectronics is utilized, enabling adjusting of transistors’ threshold voltages ($V_{TH}$). In addition, a very wide $V_{TH}$ tuning range of ∼250 mV for FD-SOI versus ∼10 mV for bulk CMOS, very good analog intrinsic performances, high-performance energy-efficient solutions, negligible input current, and reduced parasitic capacitances are achieved [15]. As a result, all transistors in the circuit operate in the strong inversion region. The proposed AFE uses a fully differential difference amplifier (FDDA) to replace the classical three-amplifier INA topology to improve AFE parameters such as *CMRR*, power consumption, area, etc. In addition, the following PGA utilizes a fully differential structure, which subsequently enables easy implementation of a fully differential ADC. The use of a low voltage power supply together with FDSOI technology is unpublished yet in the field of AFE design for biomedical signal processing. The circuit was designed for $V_{DD}$ = 1 V, $V_{AGND}=V_{DD}/2=0.5$ V and $V_{SS}$ = 0 V.

This paper is organized as follows: Section 2 introduces the whole system at the top level. Sub-blocks such as active grounding, INA, and PGA are described in more detail in the subsections, where relevant schematics, mathematical relationships, and specific values for the utilized components are mentioned. Section 3 shows detailed transistor-level schematics of the employed amplifiers. The subsections introduce the single-ended operational amplifier (OA) and two different FDDA circuits (FDDA1 and FDDA2) designed to meet specific application requirements in the block of use. Simulations are also included in these subsections. Section 4 deals with verifying the whole system’s functionality in PVT corners (process, voltage, temperature). The achieved parameters are compared with results in similar works. Finally, Section 5 highlights the major conclusions of the work.

## 2. The Proposed Structure of the Analog Front-End (AFE)

The proposed system is designed to process all types of EMG signal sources mentioned in Table 1. This means the system can process a frequency range from 20 Hz to 2 kHz. The total gain consists of two components: a gain of the input INA and the PGA amplifier. While the gain of the INA is fixed, the PGA enables four different gains that are adjustable via a two-bit digital signal. Both active grounding and shielding methods are used to improve the system’s *CMRR* and to reduce superimposed noise and interference signals.

Replacement of a standard polysilicon resistor with a pseudo-resistor composed of two MOSFETS enables the design of an OA with a very low quiescent current. Values of these resistors are in hundreds of GΩ or even in units of TΩ [16,17,18,19]—this means currents lower than 100 pA. However, poor linearity and especially leakage currents of some devices may become significant and cause system non-functionality. For example, in the utilized technology, the leakage current of the MIM capacitor is in units of nA at 85 °C when the voltage across a capacitor is 0.1 V. Further, MOSFET´s parasitic substrate, off-state, or gate current may have a significant impact on system functionality. Therefore, polysilicon high-precision non-silicided resistors are employed in the proposed system, the block diagram of which is shown in Figure 2.

The first block contains the active ground circuit and INA, whose gain, denoted as $A_{INA}$, is set with respect to the nature of the EMG signal to ensure the INA does not saturate when the maximum signal amplitude appears at the input. It is also important to take into account the offset voltage of both the INA and active ground circuit; the sum is amplified by the gain given by the $A_{INA}$ coefficient and then may contribute to the INA saturation. The first block also contains the active ground circuit setting the common-mode voltage ($V_{CM.b}$) and shielding.

The next circuit is a passive 1st-order HPF with a cutoff frequency of 20 Hz. Its purpose is to reduce the skin potentials and motion artifacts, as mentioned earlier, and minimize the dc component arising from the previous stage. However, the low cut-off frequency requires large values of capacitors and resistors, which occupy a large silicon area. For this reason, external capacitors ($C_{HP}$) are used, while resistors ($R_{HP}$) are implemented directly on the chip. The filter´s output is led to the PGA with adjustable gains of 2, 5, 10, or 20. The total gain of the AFE can be calculated as
(1)$$A_{AFE}=A_{INA}\cdot A_{PGA}=\left\{20,50,100,200\right\}.$$

The gain of the PGA is controlled by two digital control bits decoded into one of three outputs driving particular gain settings. Then, the PGA’s output signal is converted into a digital one for further signal processing. A suitable type of ADC could be an asynchronous delta-sigma modulator presented in [2] or a continuous delta-sigma modulator [20]; both have implicit antialiasing filters, low power consumption, or relaxed requirements on analog circuits.

### 2.1. Active Grounding and Shielding

The active grounding circuit is shown in Figure 3. It consists of four operational amplifiers (OA${}_{1-4}$), five resistors, and two capacitors. Two unity gain stages A${}_{1}$ and A${}_{2}$ prevent biopotential signals from being affected by load currents produced by resistors in a common-mode voltage ($V_{CM,b}$) generator comprising two resistors ($R_{CM}$) with a value of 500 kΩ. The following unity gain stage (A${}_{3}$) output is connected to an inverting A${}_{4}$ configuration and a low-impedance conductor (shield), which surrounds the input signal line and prevents external interference. Shielding brings several benefits, such as minimizing the effect of unequal cable capacities or minimizing leakage currents [21,22,23].

With a feedback loop gain of $A_{F}$, the effective impedance of E3 is significantly reduced. This scheme is also known as driven-right-leg to suppress 50/60 Hz interference [24]. When active grounding is employed, $V_{CM,b}$ is defined as
(2)$$V_{CM,b}=\frac{Z_{RL}I_{L}}{1+R_{F}/R_{I}},$$
where $I_{L}$ is the displacement current from the mains with a value in the range of few nA to several μA depending on the surroundings, and $Z_{RL}$ is the active grounding electrode impedance, which is reduced by a gain of $A_{F}$, mathematically described as
(3)$$A_{F}=1+R_{F}/R_{I}.$$

Increasing $A_{F}$ leads to a decrease in the $V_{CM,b}$ deviation. Resistor values are designed to give $A_{F}$ = 40 and simultaneously low A${}_{4}$ loading current, which results in a reduction in power consumption of A${}_{4}$. Therefore, $R_{I}$ = 20 kΩ and $R_{F}$ = 780 kΩ have been chosen. Resistor $R_{0}$ does not affect $V_{CM,b}$. Its function is associated with protecting a patient from a possible excessive flow of currents due to the A${}_{4}$ defect when this resistor limits A${}_{4}$ output current to the maximal value given by $V_{DD}/R_{0}$. The output current is limited to 50 µA in this design, thus $R_{0}=20\;k\Omega$.

Additionally, the amplifiers’ offset or polarization voltage across the electrode interface may introduce further deviation in $V_{CM,b}$, increasing the amplifiers’ input dynamic range requirements. Considering these parameters, (Equation 2) can be extended; thus
(4)$$V_{CM,b}=\frac{Z_{RL}I_{L}}{1+R_{F}/R_{I}}+\frac{V_{O4}-V_{O3}}{2\left(1+R_{I}/R_{F}\right)}+\frac{V_{P1}+V_{P2}+V_{P3}+V_{O1}+V_{O2}+V_{O4}}{2\left(1+R_{F}/R_{I}\right)}.$$
where $V_{O1-4}$ denotes the offset voltages of amplifiers A${}_{1-4}$ from Figure 3; $V_{P1-3}$ are the polarization voltages of electrodes. For simplicity, it is assumed that $R_{CMA}=R_{CMB}$ (without mismatch errors).

Due to the active grounding circuit composing negative feedback, the loop stability must be examined. Capacitors $C_{1}$ = 10 pF and $C_{F}$ = 190 pF are implemented to provide a stable active ground loop, which requires a phase margin of at least 45° across PVT corners [25].

### 2.2. Instrumentation Amplifier (INA)

The input amplifier stage is realized via an INA with a high *CMRR* for interference reduction. However, a classical triple operational amplifier INA’s *CMRR* strongly depends on a mismatch of internal resistances, where perfect matching is necessary to obtain a high *CMRR*. If a triple-OA INA is utilized, an HPF must be added to the INA input to reduce the dc offset. Detailed analysis of different INA structures is given in [26,27]. In this paper, an FDDA INA architecture has been chosen. The FDDA architecture can obtain high *CMRR* without perfect matching of resistances; it has good *CMRR* even if there are variations in process parameters. The architecture offers good performance in high input impedance and high *CMRR*. The maximum gain is limited so as not to make the DDA output saturated to the supply voltage. According to Table 1, the maximum input signal amplitude is $V_{sig}$ = 10 mV. Furthermore, if $V_{P1-3}$ = 220 mV [27,28], $V_{O1-4}$ = 5 mV, $Z_{RL}$ = 51 kΩ, and $I_{L}=5$ µA are considered, the theoretical maximum voltage swing at A${}_{1,2}$ input can be calculated from
(5)$$\begin{array}{cc} \Delta V_{A1,2} & =V_{CM}\pm \;\left(\left|\Delta V_{CMA1,2}\right|+\left|V_{sig}\right|\right)\\ & =V_{CM}\pm \;\left[\frac{Z_{RL}I_{L}}{1+R_{F}/R_{I}}+\frac{\left(V_{P1}-V_{P2}-V_{O1}-V_{O2}-2V_{O3}+2V_{O4}\right)}{2\left(1+R_{I}/R_{F}\right)}+\frac{V_{P1}+V_{P3}+V_{OF4}}{1+R_{F}/R_{I}}+\frac{V_{sig}\left(R_{F}+2R_{I}\right)}{2\left(R_{F}+R_{I}\right)}\right]. \end{array}$$

By assuming that $V_{O1}=V_{O2}=V_{O3}=V_{O4}=V_{O}$, $V_{P1}=V_{P2}$ and $R_{F}\gg R_{I}$, (Equation 5) can be simplified to
(6)$$\Delta V_{A1,2}=V_{CM}\pm \;\left[\frac{Z_{RL}I_{L}}{A_{F}}-\frac{2V_{O}}{2}+\frac{V_{P1}+V_{P3}+V_{OF4}}{A_{F}}+\frac{V_{sig}}{2}\right],$$
which results in $\Delta V_{A1,2}=V_{CM}\;\pm \;28$ mV in the worst case; based on this value, the closed-loop gain ($A_{INA}$) is set to 10. Parameter $A_{INA}$ is defined as
(7)$$A_{INA}=\frac{A_{ol}}{1+A_{ol}\frac{R_{1}/2}{R_{1}/2+R_{2}}}≅1+\frac{2R_{2}}{R_{1}},$$
where $A_{ol}$ is the DDA’s open-loop gain, $R_{1}$ = 100 kΩ and $R_{2}$ = 450 kΩ.

The next important property of the INA is its high input impedance. Large electrode impedance may attenuate the biopotential signal before amplification due to the limited INA input impedance ($R_{IN}$). Moreover, a low-frequency biopotential signal near dc is prone to distortion since the capacitance $C_{C}$ reduces the electrode impedance as the frequency increases. If the impedance is not high enough, the phase shift of the low-frequency signal can be pronounced [27,29].

### 2.3. High-Pass Filter (HPF) and Programmable Gain Amplifier (PGA)

Before further amplifying, the dc offset of the PGA´s input signal must be removed. The classical HPF with resistors connected to ground suffers from finite common-mode input impedance that may degrade the *CMRR*. A better option is to utilize a floating HPF, which has very large common-mode input impedance achieved by eliminating resistors to ground while the cutoff frequency is still the same. Component parameters are $R_{HP}$ = 800 kΩ and $C_{HP}$ = 10 nF, for a cutoff frequency of 20 Hz, when $C_{HP}$ is implemented as the external component. As will be proposed in the next section, the FDDA1 uses a CMFB circuit that includes a resistor divider composed of two resistors, designated $R_{CM,A}$ and $R_{CM,n}$. These resistors are also utilized in the HPF to save power and chip area, as illustrated in Figure 4.

The topology of the PGA is the same as the INA circuit, with the exception of an adjustable $R_{1}$ value consisting of a chain of resistors $R_{G1}$ = 26.3 kΩ, $R_{G2}$ = 29.2 kΩ, $R_{G3}$ = 194.5 kΩ, and $R_{G4}$ = 250 kΩ in series. Three CMOS switches set the $R_{1}$ value to 52.6 kΩ, 111 kΩ, 500 kΩ, or 1 MΩ between the PN and NP inputs of the FDDA2. Minimum gain is set as the default condition in the case of all switches being OFF to prevent glitches during switching among measured ranges. Resistor $R_{G5}$, with a value of 500 kΩ, is divided into 50 kΩ segments to increase the resistor’s matching accuracy.

### 2.4. Noise

Amplifiers A${}_{1-4}$ generate noise coupled to the active ground signal line. However, this signal is applied to both electrodes E1 and E2, and due to fully differential signal processing, the noise signal of A${}_{3,4}$ is canceled out. For the cascade of gain stages, the most important noise contribution comes from the first stage. To represent the noise of each stage with an input referred generator, the AFE´s noise is given by
(8)$$\bar{v_{n,out}^{2}}={\left(A_{INA}A_{PGA}\right)}^{2}\;\bar{v_{n,PGA}^{2}}+A_{PGA}\;\bar{v_{n,PGA}^{2}}.$$

Referring to the input of the AFE, this results in
(9)$$\bar{v_{n,in}^{2}}=\bar{v_{n,INA}^{2}}+\frac{\bar{v_{n,PGA}^{2}}}{A_{INA}^{2}}.$$

This proves the assumption that the noise of the first stage is the most significant. The same is valid for an internal topology of multi-stage operational amplifiers, where the noise of the first stage implemented as a differential pair is dominant.

## 3. Circuit Design on the Transistor Level

In this section, the implementation of the three different amplifiers at the transistor level will be described. The amplifiers utilize the band-gap reference (IP core), which provides the $V_{CM}$ and multiple current outputs $I_{BIASP}$ and $I_{BIASN}$. The circuit power supply is 1 V ($V_{DD}$ = 1 V). This $V_{DD}$ does not provide sufficient voltage headroom for cascaded topologies if all MOSFETs work in strong or moderate inversion. The employed technology offers regular (RVT) and low-voltage threshold (LVT) MOSFETs. However, they cannot be easily mixed without guard ring protection in some cases.

### 3.1. Operational Amplifier (A${}_{1}$–A${}_{4}$)

The folded cascode has been chosen as the structure of the amplifier A${}_{1-4}$ mainly due to the high gain–bandwidth product (*GBW*); it has only one single non-dominant pole, and it is very fast. The folded cascode structure is also an excellent first stage for a two-stage Miller CMOS OTA [30]. The topology of the circuit utilized in the components $A_{1-4}$ is shown in Figure 5.

An important parameter of the amplifier is the input common mode range ($V_{ICMR}$). As has been derived from (Equation 6), the maximum voltage swing at A${}_{1,2}$ input is $V_{CM}\;\pm \;28$ mV in the worst case. Next, the amplifier A${}_{3}$ is connected to the $V_{CM,b}$ node; its input voltage range is not affected by $V_{sig}$ and is directly equal to the $\Delta \;V_{CM,b}$ variation according to (Equation 4). That is, ±18 mV. The last amplifier (A${}_{4}$) used in an inverting configuration keeps the input voltage at noninverting and inverting terminals at $V_{CM}$ and $V_{CM}+V_{O4}$, respectively. Based on these considerations, the input differential pair is designed with LVT PMOS transistors (M${}_{1-2}$) and their threshold voltage ($V_{TH}$) is further reduced by connecting the back gate electrode to $V_{SS}$. Besides the input transistors, the threshold voltages of RVT transistors M${}_{8,9,11}$ are also lowered by connecting the back gate electrode to $V_{SS}$. In the case of M${}_{6,7}$ (LVT NMOS), their back gate electrode is connected to the transistors’ sources only to eliminate the body effect. Care must be taken to ensure sufficient voltage remains on the M${}_{3,4}$ drains and that they operate in the saturation region. Mathematically, $V_{DS3,4}$ is
(10)$$V_{DS3,4}=V_{B}-V_{GS6,7}=V_{B}-V_{TH6,7}-V_{OV6,7}.$$

Voltage $V_{GS6,7}$ is set by $V_{B}$, whose maximum value is limited by the voltage headroom in the branch M${}_{13,15,16}$. When $V_{TH}$ variation is considered as well as the body effect occurring if the back gate electrode is connected to $V_{SS}$, transistor M${}_{13}$ saturation would not be guaranteed.

#### 3.1.1. Input Common Mode Range

The minimum value of $V_{ICMR}$ of the proposed folded cascode topology is limited by the requirement that M${}_{1,2}$ operate in saturation at all times. Thus, $V_{ICMmin}$ should be at most $V_{TH}$ below the voltage at the drains of M${}_{1,2}$. The latter voltage is determined by $V_{B}$ and must allow for a voltage drop across M${}_{3,4}$ at least equal to their overdrive voltage, $\left|V_{OV3}\right|=\left|V_{OV4}\right|$. Assuming that M${}_{3,4}$ are operated at the edge of saturation, $V_{ICMmin}$ is
(11)$$V_{ICMmin}=V_{SS}+\left|V_{OV3}\right|-V_{TH1}.$$

The maximum value of $V_{ICM}$ is limited by the need to keep M${}_{5}$ always operating in saturation, which is assured by keeping the voltage across M${}_{5}$ equal to or greater than $V_{OV5}$ at all times. Thus
(12)$$V_{ICMmax}=V_{DD}-\left|V_{OV5}\right|-V_{TH1}-\left|V_{OV1}\right|.$$

While the minimum value can be easily achieved, the maximum value needs to design M${}_{1,2}$ and M${}_{4,5}$ with the minimum possible $V_{OV}$ and use LVT transistors instead of RVT in the case of M${}_{1,2}$. Furthermore, $V_{TH}$ of M${}_{1,2}$ is further lowered by connecting the back gate electrode to $V_{SS}$.

Due to the fact that A${}_{1-3}$ are connected as voltage followers, their output voltage swing is the same as the input voltage range. The inverting amplifiers’ output is directly connected to the active ground electrode; thus, A4’s output voltage swing should be as large as possible.

#### 3.1.2. Voltage Gain

The second stage is added to the circuit to achieve a higher voltage gain, higher output voltage swing, and the capability to drive the resistance load without a major impact on voltage gain and phase margin. Further, when the output voltage is close to $V_{DD}$, the output transistor M${}_{11}$ enters the linear region, and the second stage loses gain, there is still sufficient gain remaining in the first stage to suppress the distortion. The dc open-loop gain can be mathematically expressed as
(13)$$A_{V0}=g_{m1}g_{m11}\left\{\left[g_{m6}r_{ds6}\left(r_{ds2}\|r_{ds4}\right)\right]\|r_{ds8}\right\}\left(r_{ds10}\|r_{ds11}\right).$$

#### 3.1.3. Noise

The noise analysis is based on the assumption, as mentioned earlier, that the cascoding amplifier stages have a minor impact on the equivalent input-referred noise because their contribution is attenuated by the voltage gain of the previous stage. The same situation applies in a two-stage amplifier for the transistors in the second stage.

The noise of the current source implemented via M${}_{5}$ has a negligible effect on the output voltage since it flows evenly in the two branches, which are assumed to be well-matched. The noise of cascaded transistors M${}_{6,7}$ is also neglected because their noise gain to the output is considerably lower than the gain from M${}_{1,2}$, M${}_{3,4}$, and M${}_{8,9}$. The contribution of the transistors M${}_{1,2}$, M${}_{3,4}$, and M${}_{8,9}$ to the input-referred noise voltage can be expressed as $\bar{i_{n1}^{2}}/g_{m1}^{2}$, $\bar{i_{n3}^{2}}/g_{m1}^{2}$, and $\bar{i_{n8}^{2}}/g_{m1}^{2}$, respectively. The total input-referred noise is then given by
(14)$$\bar{v_{ni}^{2}}=\frac{2\;\left(\bar{i_{n1}^{2}}+\bar{i_{n3}^{2}}+\bar{i_{n8}^{2}}\right)}{g_{m1}^{2}},$$
where
(15)$$\bar{i_{nj}^{2}}=\bar{i_{th,j}^{2}}+\bar{i_{1/f,j}^{2}}=4kT\gamma g_{m}+\frac{K_{F}}{W_{j}L_{j}C_{ox}}\frac{g_{m,j}^{2}}{f},$$
where *j* denotes the number of the transistor (1, 3, or 8), *k* is Boltzmann’s constant, *T* is the temperature in kelvin, γ is the bias-dependent noise parameter, $g_{m}$ is the transistor’s transconductance, $K_{F}$ is the flicker noise coefficient (process-dependent constant), $C_{ox}$ represents the oxide capacitance per unit area, *L* is the transistor length, $I_{D}$ is the drain current, and *f* is the frequency. Note that a bandwidth of 1 Hz is assumed.

The flicker (or 1/f) noise corner frequency is
(16)$$f_{c}=\frac{K_{F}}{\gamma W_{j}L_{j}C_{ox}}\frac{g_{m,j}}{4kT}.$$

This result implies that $f_{c}$ generally depends on the device’s area and transconductance. To reduce the impact of 1/f noise, the device’s area must be large. Generally, PMOS devices exhibit less 1/f noise than NMOS transistors.

The formula for calculating the total thermal noise and 1/f noise of a single transistor for a band from $f_{1}$ to $f_{2}$ can be obtained by modifying (Equation 14). That is,
(17)$$\begin{array}{c} \bar{i_{n(f_{1}-f_{2}),j}^{2}}=4kT\gamma g_{m,j}\left(f_{2}-f_{1}\right)+\frac{K_{F}g_{m,j}^{2}}{W_{j}L_{j}C_{ox}}\int_{f_{1}}^{f_{2}}\frac{df}{f}\approx 4kT\gamma g_{m,j}f_{2}+\frac{K_{F}g_{m,j}^{2}}{W_{j}L_{j}C_{ox}}\ln\left(f_{1}\right). \end{array}$$

Simplification in the second part of (Equation 17) can be performed if $f_{2}\gg f_{1}$. At frequencies beyond $f_{c}$, the contribution of 1/f noise can be ignored, while the thermal noise becomes dominant.

#### 3.1.4. Simulations

The frequency responses of the amplifier are represented in Figure 6. Figure 6a shows that the input-referred noise varies from 2.5 to 3.2 µV/$\sqrt{Hz}$ at 1 Hz over the process corners. The total integrated input-referred noise in the frequency range 20 Hz–2 kHz is obtained; 7.6 µV in the typical process corner. Next, as presented in Figure 6b, the input impedance corner values are 26.5 GΩ (FF corner) and 48 GΩ (SS corner) at the frequency of 1 Hz. The value for the typical process is 14.2 GΩ. Figure 6c indicates that the dc gain is in the range 118 dB–132 dB, with a value of 128 dB (phase margin of 70°) for the typical process corner. The results also demonstrate that the proposed amplifier is stable (phase margin > 61° in the worst case). Finally, Figure 6d shows *CMRR* = 88 dB in the typical process corner when it drops to 78 dB in the worst case. The standard deviation of the dc offset is 2.1 mV, and the simulated power consumption is 8 µW.

### 3.2. Fully Differential Difference Differential Amplifier (FDDA1)

The gain stage is composed of an amplifier, denoted as FDDA1, which is implemented as a fully differential non-inverting gain stage with a very large input impedance. Due to the fully differential structure, designing a common-mode feedback (CMFB) circuit was also required. The CMFB loop could adjust either $I_{5,6}$ or $I_{7,8}$ by controlling the gate voltages of the transistors that generate those currents. The CMFB’s gain is provided by common-source NMOS transistors M${}_{7,8}$, which have larger $g_{m}$ than PMOS transistors when the same size is used. The output of FDDA1 is loaded by resistors in the CMFB circuit, resistors in the feedback loop setting the gain ($A_{INA}$), and components in the HPF comprising further resistors and large external capacitors. Therefore, the amplifier is designed with class AB output stages to ensure the capability to drive this load. Moreover, they can generate much greater output currents than the total bias current in the output stage and save power. The complete circuit schematic is shown in Figure 7.

The input differential pairs consist of LVT PMOS transistors M${}_{1-2}$ and M${}_{3-4}$ with the back gate electrode connected to $V_{SS}$. Then, the input stage provides a sufficiently large input voltage range when expected fluctuation has been derived in (Equation 6) to be $V_{CM}\;\pm$ 28 mV. Three inverting common-source amplifiers follow the differential stage. The first two inverting amplifiers (M${}_{9,11}$ or M${}_{10,12}$ and M${}_{13,15}$ or M${}_{14,16}$) are combined with the gate-drain feedback to behave as a single noninverting common-source stage. The gain-drain feedback is created across the second common-source amplifier by resistor $R_{G}$. Transistors M${}_{13,20}$ and M${}_{17,18}$ form the output class AB rail-to-rail stage. A detailed analysis of this three-stage class AB topology has been proposed in [31].

#### 3.2.1. Input Common-Mode Range

The minimum value of $V_{ICMR}$ of the proposed FDDA1 topology is limited by the requirement that M${}_{1-4}$ operate in saturation at all times. Thus, $V_{ICMmin}$ should be at most $V_{TH}$ below the voltage at the drains of M${}_{1-4}$. Assuming that M${}_{7,8}$ are operated at the edge of saturation, $V_{ICMmin}$ is given by
(18)$$V_{ICMmin}=V_{SS}+\left|V_{OV7}\right|-V_{TH1}.$$

Due to the fact that FDDA1 operates in the closed-loop configuration with a gain of 10, the output voltage can vary over a range determined by (Equation 6) extended with the FDDA1 offset. Thus,
(19)$$\Delta V_{OP}=\Delta V_{ON}=V_{CM}\pm \;\left[\frac{Z_{RL}I_{L}}{A_{F}}-\frac{2V_{O}}{2}+\frac{V_{P1}+V_{P3}+V_{OF4}}{A_{F}}+\frac{V_{sig}}{2}+V_{OF1}\right],$$
where $V_{OF1}$ is FDDA1´s offset voltage. According to (Equation 6), the output voltage swing can reach values up to $V_{CM}\pm \;330\;mV$ when $V_{OF1}$ = 5 mV is assumed.

The maximum value of $V_{ICM}$ is limited by the need to keep current mirrors composed of M${}_{5,6}$ always operating in saturation, which is assured by keeping the voltage across M${}_{5,6}$ equal to or greater than $V_{OV5,6}$ at all times. Then,
(20)$$V_{ICMmax}=V_{DD}-\left|V_{OV5}\right|-V_{TH1}-\left|V_{OV1}\right|.$$

#### 3.2.2. Voltage Gain

The overall gain and effective number of amplifier stages are decreased from four to three by introducing feedback resistor $R_{G}$. The first stage consists of differential pairs with the active load formed by M${}_{1-8}$. Its gain can be expressed as
(21)$$A_{S1}=g_{m1}\left(r_{ds1}\left\|r_{ds3}\right\|r_{ds7}\right).$$

The second amplifier stage is constituted from the cascade of the first two inverting common-source amplifiers (M${}_{9-12}$ and M${}_{13-16}$) that behave as a single stage. The second stage gain is given as [31]
(22)$$A_{S2}^{*}=A_{S2}\;A_{S3}\approx g_{m9}R_{G}.$$

The gain of the amplifier is approximately the product of the gains of all three stages and is given by
(23)$$A_{FDDA1}=A_{S1}\;\left[A_{S2}^{*}\;g_{m19}\left(r_{ds17}\|r_{ds19}\right)+g_{m17}\left(r_{ds17}\|r_{ds19}\right)\right].$$

#### 3.2.3. Noise

The transistors in the amplifier’s first stage have a dominant effect on the FDDA1 noise, as was mentioned for multistage amplifiers in Section 2.4. The noise of the current sources implemented via M${}_{5,6}$ have a negligible effect on the output voltage since they affect evenly all four branches, which are assumed to be well-matched. All other transistors in the first stage affect the input-referred noise as follows
(24)$$\bar{v_{ni}^{2}}=\frac{2\bar{i_{n1}^{2}}+\bar{i_{n7}^{2}}}{g_{m1}^{2}},$$
where $\bar{i_{n,j}}$ (*j* denotes the transistor number) has been defined in (Equation 15).

Since the amplifier operates with resistor feedback, it is necessary to consider their noise. Figure 8 shows the implementation of a feedback amplifier using FDDA1, including noise sources modeling the resistors’ thermal noise.

Each source makes its own contribution to the noise at the amplifier’s output. At the beginning, input-referred noise is derived. Resistor R${}_{1}$ is divided into two parts, R${}_{1A}$ and R${}_{1B}$; each belongs to the corresponding feedback. A node between R${}_{1A}$ and R${}_{1B}$ behaves as the virtual ground; that is, $V_{CM}$ in this case. Then, each feedback path can be analyzed separately. Firstly, the impact of path A on the amplifier input-referred noise can be mathematically described as
(25)$$\bar{v_{niA}^{2}}=\sqrt{{\left(\frac{R_{2A}}{R_{1A}+R_{2A}}\;\bar{v_{nR1A}}\right)}^{2}+{\left(\frac{R_{1A}}{R_{1A}+R_{2A}}\;\bar{v_{nR2A}}\right)}^{2}+\bar{v_{ni}^{2}}},$$
where the noise from a resistor is modeled as $\bar{v_{nR}}=\sqrt{4kTR}$.

Extending the noise analysis to the feedback path B, assumed as well-matched with the feedback path A, an equation for the overall FDDA1 input-referred noise can be written as
(26)$$\bar{v_{niAB}^{2}}=\sqrt{\bar{v_{niA}^{2}}+\bar{v_{niB}^{2}}}=\sqrt{2\;\bar{v_{niA}^{2}}}=\sqrt{2\;\bar{v_{niB}^{2}}}.$$

The output-referred noise can be simply calculated by multiplying the input-referred noise by the gain of the amplifier.
(27)$$\bar{v_{noAB}^{2}}=\frac{R_{2A}}{R_{2A}+R_{1A}}\;\bar{v_{niAB}^{2}}.$$

It should be noted that a single-pole system is assumed where the feedback network is purely resistive and that the noise gain versus frequency is flat. Otherwise, capacitances (e.g., FDDA1’s input capacitance, load capacitance, etc.) cause the noise gain not to be constant over the bandwidth of interest, and more complex techniques must be used to calculate the total noise.

#### 3.2.4. Simulations

The simulation results for FDDA1 are presented in Figure 9. Figure 9a shows that the input-referred noise of FDDA1 is 1.6 µV/$\sqrt{Hz}$ at 1 Hz over the process corners, with the resistors in the feedback having a negligible effect on the overall noise. The total integrated input-referred noise in the frequency range 20 Hz–2 kHz is obtained as 5.7 µV. Next, as presented in Figure 9b, the input impedance corner values are 8.6 GΩ (FF corner) and 54 GΩ (SS corner) at the frequency of 20 Hz. The value for the typical process is 37 GΩ. Figure 9c indicates that the gain varies from 91.1 dB to 101.5 dB, with a value of 98.7 dB (phase margin of 80.6°) for the typical process corner. The results also demonstrate that the proposed system is stable (phase margin > 78.5° in the worst case). The *CMRR* parameter of the fully differential FDDA1 was verified using Monte Carlo analyses that comprised fifth-hundred samples for the global and local process variations. Figure 9d shows the mean value of 89 dB when the sigma deviation is 9.42 dB. The standard deviation of the dc offset is 3.9 mV, and the simulated power consumption is 7.1 µW.

### 3.3. Fully Differential Difference Amplifier for PGA (FDDA2)

The signal coming into the PGA block is divested of the dc component, and thereby, FDDA2 amplifies only the ac signal in the bandwidth of interest together with a small dc component in the form of its own offset. The maximum amplitude of the ac signal from FDDA1 is given by $V_{sig}\cdot A_{INA}$ = 100 mV. Considering the impact of process variations, it would not be possible for a simple input differential pair to process such amplitude; hence the differential pair is designed as rail-to-rail with two input pairs (PMOS and NMOS) in parallel as illustrated in Figure 10.

Both input differential pairs utilize LVT transistors with further lowered $V_{TH}$ by connecting their back gate electrode $V_{DD}$ and $V_{SS}$ in the NMOS and PMOS pair, respectively. However, this solution needs to create triple-well isolation in the case of the PMOS transistors, negatively affecting the input voltage offset.

For $V_{ICMRmin}$ and $V_{ICMRmax}$, we can write
(28)$$V_{ICMmin}=V_{SS}+\left|V_{OV15}\right|-V_{TH1P},$$
(29)$$V_{ICMmax}=V_{DD}-\left|V_{OV9}\right|-V_{TH1N}.$$

Since the amplifier’s output is loaded by a large resistance composed of resistors in the CMFB circuit and feedback, the second stage employs a rail-to-rail class A common source amplifier. Then, the dc open-loop gain is given by
(30)$$A_{FDDA2}=\left(g_{m1P}+g_{m1N}\right)\left\{\left[g_{m14}r_{ds14}\left(r_{ds1P}\|r_{ds16}\right)\right]\|\left[g_{m12}r_{ds12}\left(r_{ds1N}\|r_{ds10}\right)\right]\right\}g_{m18}\left(r_{ds18}\|r_{ds20}\right).$$

The noise is derived according to the principles mentioned for the previous two amplifiers. Then, we can write
(31)$$\bar{v_{niPGA}^{2}}=\frac{2\bar{i_{nP1}^{2}}+\bar{i_{n16}^{2}}+2\bar{i_{nN1}^{2}}+\bar{i_{n10}^{2}}}{g_{mP1}+g_{mN1}},$$
where $\bar{i_{n,j}}$ (*j* denotes the transistor number) has been defined in (Equation 15).

If the noise of the whole PGA is considered, i.e., with feedback resistors, the total noise will depend on the selected gain. Equation (Equation 27) can therefore be rewritten as
(32)$$\bar{v_{noPGA}^{2}}=\frac{R_{G2}}{R_{G2}+\sum_{k=2}^{20}R_{G1n}}\;\bar{v_{niPGA}^{2}},$$
where *k* can be an even number in the specified interval.

The simulation results for FDDA2 are presented in Figure 11. Figure 11a shows that the input-referred noise of FDDA2 is 3 µV/$\sqrt{Hz}$ at 1 Hz over the process corners, with the resistors in the feedback having a negligible effect on the overall noise. The total integrated input-referred noise in the frequency range 20 Hz–2 kHz is obtained as 7.5 µV. Next, as presented in Figure 11b, the input impedance corner values are 7.4 GΩ (FF corner) and 61 GΩ (SS corner) at the frequency of 20 Hz. The value for the typical process is 35 GΩ. Figure 11c indicates that the gain varies from 97.6 dB to 120.6 dB, with a value of 113 dB (phase margin of 80.3°) for the typical process corner. The results also demonstrate that the proposed system is stable (phase margin > 79° in the worst case). The *CMRR* parameter of the fully differential FDDA1 was verified using Monte Carlo analyses that comprised fifth-hundred samples for the global and local process variations. Figure 11d shows the mean value of 75.9 dB when the sigma deviation is 9.42 dB. The standard deviation of the dc offset is 4.4 mV, and the simulated power consumption is 8.6 µW.

## 4. Simulation Results

The proposed AFE was designed using STMicroelectronics 28 nm FDSOI technology with a 1 V supply voltage. Proper functionality against PVT variations was verified; a supply voltage variation of $V_{DD}\pm 10\%$ and a commercial temperature range from 0 °C to 70 °C were considered. The circuit layout has not been created; nevertheless, an estimate of the area is presented, derived from the sum of the areas occupied by each component, setting according to the boundary layer with a margin for metal paths. The total area is 0.09${}^{2}$ mm, with the amplifiers (A${}_{1}$–A${}_{4}$) occupying 0.017 mm${}^{2}$ (OA), 0.021 mm${}^{2}$ (FDDA1), 0.008 mm${}^{2}$ (FDDA2), and 0.034 mm${}^{2}$$C_{F}$, together with $C_{1}$, and the rest of the area is occupied by feedback resistors.

In the first step, the active ground loop stability was investigated. The electrical model of the Ag/AgCl electrode with component values used for the simulation was adopted from [32]. The simulation results are shown in Figure 12, including the circuit diagram of the electrode with the component values in Figure 12a. Figure 12b shows the extracted phase margin of all PVT corner runs. In order to secure the active ground feedback loop’s stability, two capacitors, $C_{1}$ = 10 pF and $C_{L}$ = 190 pF, are implemented into the circuit. As can be seen from Figure 12b, the minimum value of the phase margin over the PVT range is 44.1°. For typical conditions, the phase margin takes the value 54.7°. The gain of the feedback loop is 32 dB, which corresponds to the assumptions in (Equation 3).

In the second step, the *CMRR* of the system was analyzed. The input signal source was connected between electrodes E2 and E3, while electrodes E1 and E2 were connected together; thus, the same signal occurred at two FDDA1 inputs. The impact of *CMRR* was probed at the fully differential output of FDDA2. In order to find correct values, the almost ideal symmetry of the fully differential structure must be broken. Therefore, a Monte Carlo analysis with 500 runs was employed, where both process and mismatch variation were performed. As a result, the frequency response and several histograms are shown; the mean value and standard deviation of 1 σ are noted for each of them, as shown in Figure 13. The results show that the mean value of *CMRR* fluctuates from 92.5 to 111.5 dB (105.5 typical) and a standard deviation of about σ≈10.5 dB is obtained.

Figure 14a depicts the input impedance of the system, which consists of a parallel combination of input resistance OA and FDDA1. The typical dc value of 11 GOhm is achieved, while the minimum dc value from the PVT corner analysis is 2.1 GΩ. As the resistance decreases with frequency, its minimum value, 4.2 MΩ, appears at the maximum frequency of the band of interest (2 kHz). Figure 14b illustrates the input noise of the system measured between nodes V${}_{A1}$ and V${}_{A2}$ (FDDA1 inputs), which is 3.2 µV/$\sqrt{\mathrm{Hz}}$ at 27 °C and 3.5 µV/$\sqrt{\mathrm{Hz}}$ at 70 °C in the worst case.

The system amplification for all four settings, i.e., 20, 50, 100, and 200, is shown in Figure 15a, and a table with simulated $A_{AFE}$ values at 1 kHz is also delivered. The amplification is defined by the ratio of the feedback resistors, whose values and, thereby, areas are large. The process variance will not practically affect the gain; however, a mismatch between the resistors will. Due to the large area of the resistors, good matching is achieved, and the gain error is very small in the order of tens of dB. Figure 15b shows the processing of an EMG recording [33] of a patient with neuropathy when the system gain was set to 50.

The parameters of the proposed system are summarized in Table 2, where they are further compared with related works.

Many studies are dealing with biological signal processing, mostly ECG; however, ECG signals have significantly lower bandwidths requiring a different design approach. Therefore, papers proposing systems suitable for signal processing in the order of a few kHz were chosen for the comparison table except for [37,38]. These papers are listed because the presented topology contains a similar active ground circuit and has high *CMRR* in the case of [37,38].

Compared with other works, this design presents the highest *CMRR* (105.5 dB), achieved using an active ground circuit. The active ground circuit contains several buffers; hence, higher CMRR is achieved at the cost of higher power consumption. The designed circuit is also characterized by high input impedance and tunable gain.

## 5. Conclusions

A novel circuit that processes an EMG signal in the frequency range 20 Hz–2 kHz with a tunable gain of 20, 50, 100, or 200 using advanced 28 nm FDSOI technology is proposed. A design of the transistor-level subcircuit that employs a back-gate electrode to control the threshold voltage is shown. Thanks to this solution, all transistors operate in the strong inversion region even with a supply voltage of only 0.9 V. At such a low supply voltage, an active ground circuit keeps $V_{CM}$ at the electrodes close to the analog ground voltage, which leads to a very high *CMRR*. The other parameters of the proposed system are summarized in Table 2.

The circuit performance was verified using advanced analyses (corner and Monte Carlo) of the device models provided by STMicroelectronics. Cadence Virtuoso with the Spectre simulator was used for these analyses. Although the results have not been verified on a fabricated chip, advanced analyses considering the impact of a process variation of 6 σ and a commercial temperature range of 0–70 °C credibly show the performance of the proposed circuits over full fabrication tolerances defined by the manufacturer, and hence the benefit of the proposed solution.

## Figures and Tables

**Figure 1 sensors-23-03422-f001:**
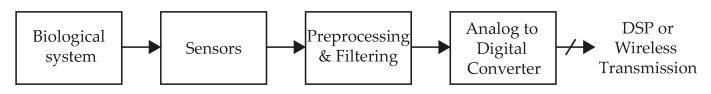
Block diagram of a typical biomedical signal processing system.

**Figure 2 sensors-23-03422-f002:**
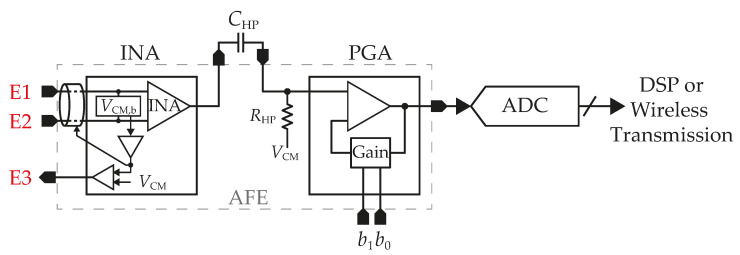
Block diagram of the proposed AFE.

**Figure 3 sensors-23-03422-f003:**
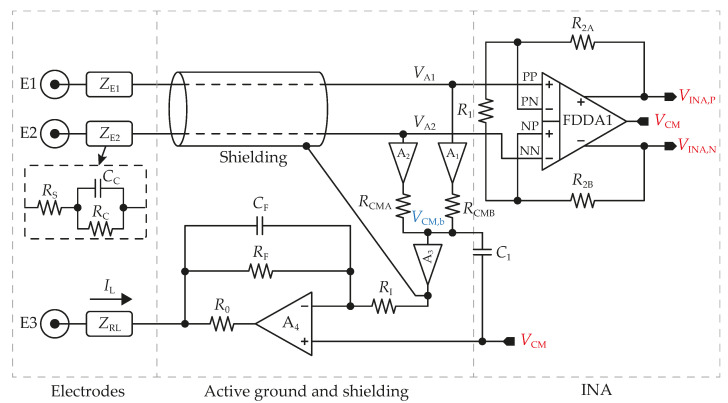
The topology of the proposed input part of the AFE includes active ground and shielding circuits.

**Figure 4 sensors-23-03422-f004:**
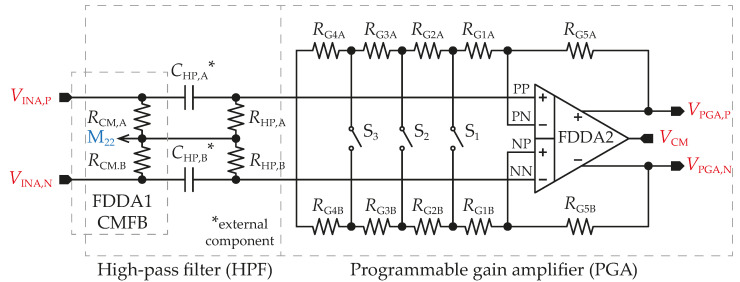
Topology of HPF and PGA.

**Figure 5 sensors-23-03422-f005:**
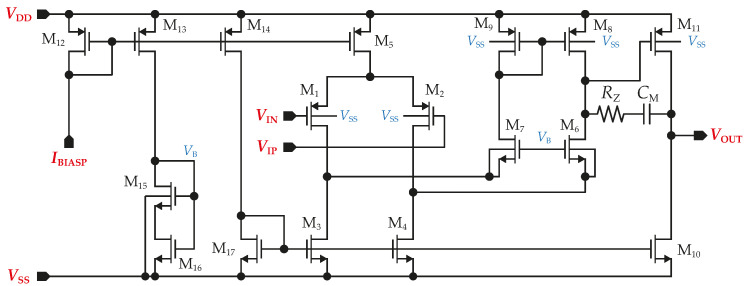
Topology of proposed operational amplifier utilized in active grounding and shielding circuitry.

**Figure 6 sensors-23-03422-f006:**
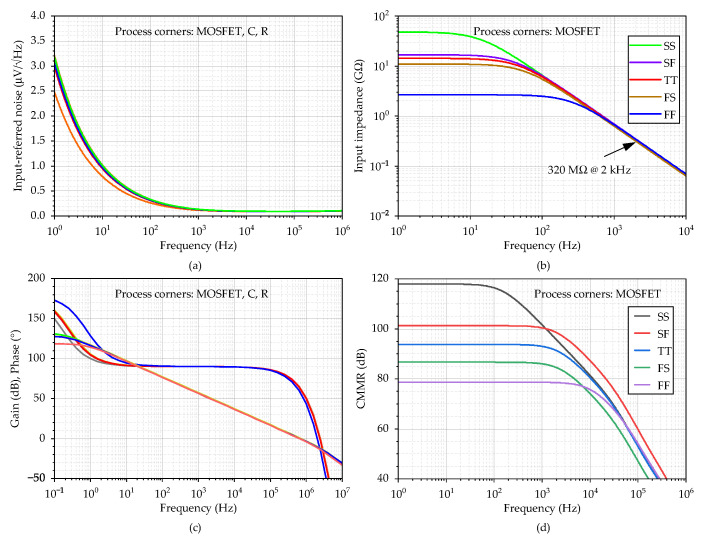
The AC analysis of the circuit from Figure 5, where (**a**) input referred noise, (**b**) input impedance, (**c**) gain including phase, and (**d**) *CMRR* are shown. The process corners of MOSFETs are listed in the order of NMOS-PMOS, where S is slow, T is typical, and F is the fast corner.

**Figure 7 sensors-23-03422-f007:**
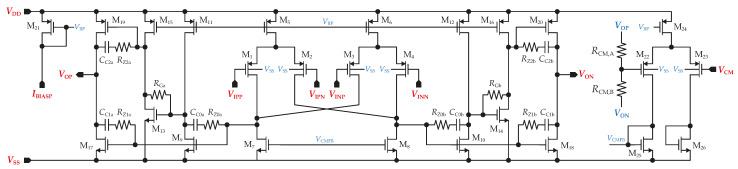
Topology of proposed FDDA1.

**Figure 8 sensors-23-03422-f008:**
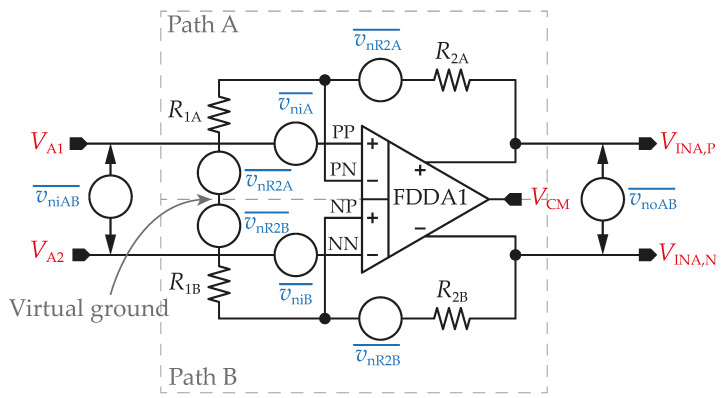
FDDA1 noise model.

**Figure 9 sensors-23-03422-f009:**
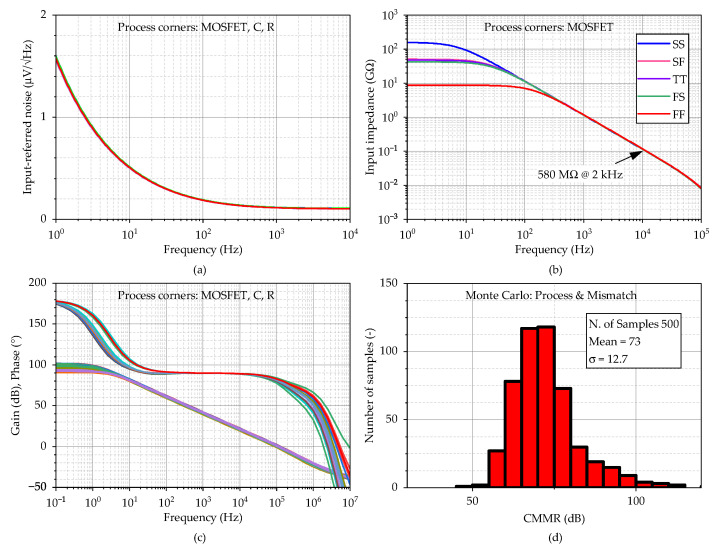
AC analysis of the circuit from Figure 7, including the corners of the MOSFETs, resistors, and capacitors, where (**a**) input-referred noise, (**b**) input impedance (only MOSFET corners), (**c**) gain including phase, and (**d**) histogram of *CMRR* are shown. Process corners of MOSFETs are listed in the order of NMOS-PMOS, where S is slow, T is typical, and F is the fast corner.

**Figure 10 sensors-23-03422-f010:**
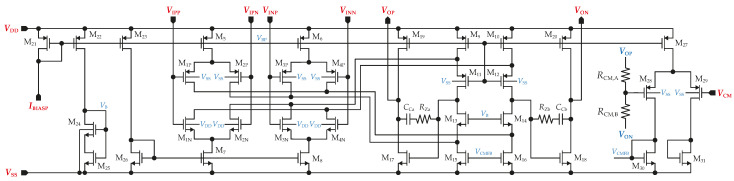
Topology of the proposed FDDA2.

**Figure 11 sensors-23-03422-f011:**
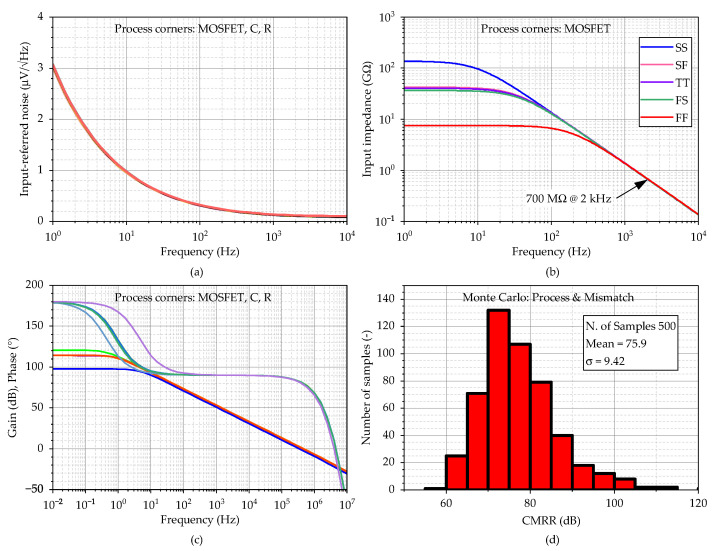
AC analysis of the circuit from Figure 10, including the corners of the MOSFETs, resistors, and capacitors where (**a**) input referred noise, (**b**) input impedance (only MOSFET corners), (**c**) gain including phase, and (**d**) histogram of *CMRR* are shown. Process corners of MOSFETs are listed in the order of NMOS-PMOS, where S is slow, T is typical, and F is the fast corner.

**Figure 12 sensors-23-03422-f012:**
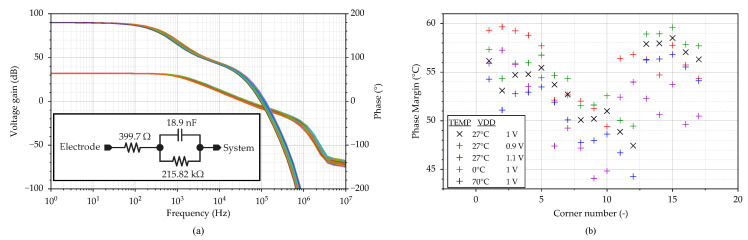
Stability analysis of the active ground loop: (**a**) gain and phase waveform and (**b**) values of phase margin over PVT corners.

**Figure 13 sensors-23-03422-f013:**
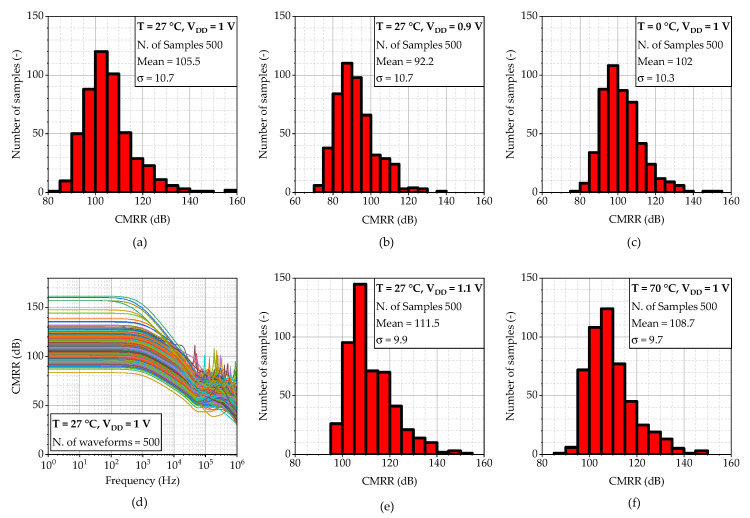
Analysis of CMRR for (**a**–**c**,**e**,**f**) different conditions by Monte Carlo and (**d**) typical AC response.

**Figure 14 sensors-23-03422-f014:**
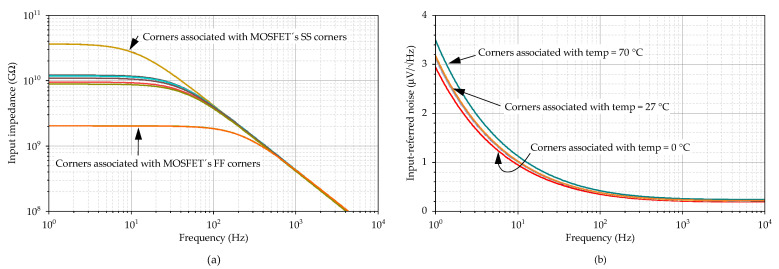
Frequency response of (**a**) the input impedance and (**b**) the input-referred noise in different PVT corners.

**Figure 15 sensors-23-03422-f015:**
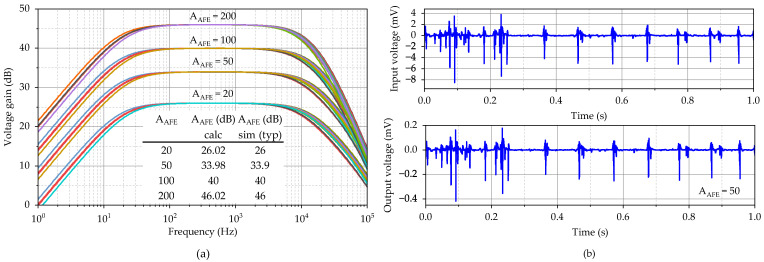
Variation in (**a**) frequency response at four different gain settings and (**b**) processing of the EMG signal.

**Table 1 sensors-23-03422-t001:** Voltage and frequency parameters of some common biopotential signals [6].

Source	Amplitude (mV)	Bandwidth (Hz)
ECG	0.5–1	0.01–150
EEG	0.002–0.1	0.5–30
EMG	0.02–10	20–2000
EOG	0.01–1	0.1–10

**Table 2 sensors-23-03422-t002:** Performance comparison table.

Parameter	2011 [34]	2017 [35]	2011 [36]	2019 [37]	2013 [38]	This Work
Power (µW)	20.8	9.9	7.92	90	8.25	48
$V_{DD}$ (V)	1.8	1.2	1.8	5	1.8	1
Gain (dB)	52.5–57.5	52	39.4	0–20	52–80	26–46
Bandwidth (Hz)	4/300–10k	1–5k	10–7.2k	0.26–100	<30 or 100	20–2k
Input noise (µV/$\sqrt{\mathrm{Hz}}$)	2.38	5	3.5	3.7	0.91	7.6
	(0.5–50k)	(10–5k)	(10–100k)	(0.5–100)	(0.5–100)	(20–2k)
CMRR (dB)	88	65	70.1	70	>90	105.5
Input impedance (GΩ)	N/A	N/A	N/A	400	>0.5	11
Area (mm${}^{2}$)	0.061	0.018	0.0625	1.23	N/A	0.09 *
Process	0.18	0.13	0.18	0.18	0.18	0.028 FDSOI

* 10 nF capacitors in the HPF are implemented externally.

## Data Availability

Not applicable.

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
