# Peer review of "A Fully Differential Analog Front-End for Signal Processing from EMG Sensor in 28 nm FDSOI Technology"

_sensors, 2023, doi:10.3390/s23073422_

Round 1

Reviewer 1 Report

Check typo errors.

For example

a) abstract: 8. row "fifth-hounded" 

b) page 8, 248. row "M1.2"

Author Response

The authors thank the reviewer for all of his valuable comments; the individual responses are outlined below.

Point 1: Check typo errors. For example,

  1. a) abstract: 8. row "fifth-hounded"
  2. b) page 8, 248. row "M1.2"

Response 1: These typos have been fixed.

Best regards

Vilem Kledrowetz

Reviewer 2 Report

The authors presented a design of an EMG front-end circuit that can be implemented in 28 nm FDSOI technology. The description of the design is provided at the transistor level, and performance is evaluated by using Monte Carlo simulations. However, there is no real novelty stated. What is new in the proposed design? What tools were used in the design and for simulations? There are no results from testing a real chip. I recommend supplementing the paper with the performance results of real integrated circuits. Table 2 shows that the power consumption is at least 2.5 times higher for this work than for other designs. Could power consumption be decreased?

Finally, is it true that STMicroelectronics prohibits the use of this 28-nm technology for any medical or military applications? https://www.cmc.ca/stm-28nm-fd-soi-cmos/

Minor comments:

1.     Line 8. Could you elaborate on what ‘fifth-hounded’ means?

2.     Line 27. Something is missing. Perhaps 'it constitutes?' It is unclear what does it constitute an analogue front-end component (AFE) together with a filter.

3.     Line 59. ‘Tents’.  You probably meant ‘tens’.

4.     Line 67. I suppose you mean ‘bulk CMOS’. The word 'bulk' has its own meaning and, therefore, cannot be searched by less experienced readers, contrary to ‘bulk CMOS‘.

5.     Line 146. ‘Existence‘? The entire sentence could be improved for clarity.

6.     Lines 152-154.  How was it examined, and how were the capacitor values selected? Is it possible to quantitatively express the sufficient phase margin and active ground loop? It is also fine if the selection was experimental, but it should be stated so.

7.     Figure 4, Line 191, Line 195. There are two pairs of R_G2A and R_G2B resistors in Figure 4 and in the following paragraph. one is 29.2k, and the other is 500k.

8.     Line 203. Choose either 'Results' or 'Resulted'?

9.     Line 217. GBW (Gain bandwidth product?) abbreviation is not stated anywhere in the work.

10.  Line 470 Scientific papers cannot be characterized by CMRR: ‘’Compared with other papers, this paper is characterized by high CMRR…’’. Please rephrase.

Author Response

The authors thank the reviewer for all of his valuable comments; the individual responses are in the attached pdf.

Reviewer 3 Report

This paper proposed a high-bandwidth and low supply voltage analog EMG sensor for biomedical signal acquisition. The amplifier circuit contains an INA and a PGA modules with fully-differential structure. The Monte Carlo simulation demonstrated the merits of the circuit design. The manuscript is well-written. 

The fonts in the figures should be revised with the MDPI font format.

The captions of Figure 9 and Figure 11 should contain the abbreviation description of SS, SF, TT, FS, and FF.

Author Response

The authors thank the reviewer for all of his valuable comments; the individual responses are outlined below.

Point 1:

The fonts in the figures should be revised with the MDPI font format.

Response 1: We revised all figures and fixed the font format in Fig. 6, Fig. 9, Fig. 11, Fig. 13, Fig. 14, Fig. 15.

Point 2:

The captions of Figure 9 and Figure 11 should contain the abbreviation description of SS, SF, TT, FS, and FF.

Response 2: We add descriptions of these corners to Fig. 6, Fig. 9, and Fig. 11.

“Process corners of MOSFETs are listed in the order of NMOS-PMOS, where S is slow, T is typical, and F is the fast corner.”

Best regards

Vilem Kledrowetz

Reviewer 4 Report

The submission “A Fully Differential Analog Front-End for signal processing from EMG Sensor in 28-nm FDSOI Technology” proposes an electromyography (EMG) signal amplifier that achieves a high gain and high input impedance at a low chip area.

Unfortunately, this represents only an incremental development in the field of electronics engineering design that lacks the novelty and potential impact needed for the publication process. It is more suitable and considerable for a technical report issue than a scientific research area.

Author Response

The authors thank the reviewer for his comment. The results of our work are difficult to compare with other works because most of them deal primarily with EEG signal processing or signals having a much lower frequency and amplitude, requiring a different approach to device design. For EMG signals processing with a supply voltage of only 1 V, we have selected FDSOI technology to achieve the requested parameters. The technology enables tunability of threshold voltages which is critical for our design. In the context of comparable publications, we have used the lowest supply voltage, bringing a remarkable profit for low-voltage battery power supply. This study presents probably the first usage of the FDSOI technology in that area. In spite of the fact that using low supply voltage makes the circuit design more challenging from the point of view of achieving a high CMRR, we have reached one of the highest CMRRs yet published due to designing a fully differential floating AFE, where matching errors are the only one, which limits CMRR. In addition, the circuit allows tuning of the AFE´s gain over a large range with a linear characteristic. With respect to other works, we can say that our contribution is comparable to other published works in all general parameters but defeats them by the low supply voltage and high tunability. These benefits were achieved through a new approach with the FDSOI technology key options.

Round 2

Reviewer 2 Report

I am happy with the final improvements to the paper.

Author Response

Dear reviewer,

Thank you for your decision.

Vilem Kledrowetz
Corresponding author

Reviewer 4 Report

In my opinion, the presented study still lake scientific novelty. The circuit simulation is not enough to present a biomedical-based design where many related criteria affect the outcome results. Maybe this study is worthwhile for other applications that require less sensitive issues.

Unfortunately, the manuscript is not suitable for publication.  

Author Response

Dear reviewer,

The presented work is based on our previous designs in EMG signal processing. The original request came from our biomedical department, where wearable patch with electronics for athletes has been developed.

The first circuit was designed with discrete components on a PCB and comprised a large box with cables. This device was fabricated and validated on athletes. However, this solution was inconvenient and unusable for a user during athletic performance, especially when multiple sensors were used. Therefore, the demand for an integrated version of the circuit arose.

However, IC implementation was difficult in standard bulk technology when using a battery power supply due to the body effect in cascodes, large leakage currents of native MOSFETs, or poor noise characteristics in the case of nA currents. Moreover, there was a problem with the fully differential pair composed of complementary pairs and its linearity; this is critical mainly in the first amplifier. We used only PMOS dif. pair in the first amplifier, which is impossible to design for the required input range and supply voltage when standard technology with standard VTH MOSFETs is used. Employing low or native VTH MOSFET is not wanted because it requires creating additional masks for the lithography process. At the same time, there is still body effect, relatively high leakage currents, or poor noise characteristics.

With modern FDSOI technology, we could meet the system's requirements. It is also possible to eliminate the body effect, and there is a wide VTH tunning range when the number of lithography masks is not increased.

The circuit implementation brings completely new possibilities for the athletes. Furthermore, it increases the athlete's comfort when measurement performance is not affected due to the unique LV design. In the future, it is possible to modify the circuit parameters via the back gate electrode driving and implement digital processing on-chip is also possible. 

Also, employing SOI technology and 1 V power supply is unique in this area.

We have many experiences in the design of analog circuits in different technologies and applications, which have been implemented on a chip and consequently have been measurements implemented in a target application. CAD simulations' results (verified by MC or corner sim.) closely match the fabricated device measurements and reliably demonstrate the circuits' functionality and benefits. Also, LETI-UTSOI surface-potential-based model that STM 28 nm design kit uses is very accurate and reliable.

We want to present the results to our potential partners via this paper. With their support, we can move forward to produce and test the chip under real conditions.

Yours sincerely

The authors' team